# Genetic Evidence of *Yersinia pestis* from the First Pandemic

**DOI:** 10.3390/genes16080926

**Published:** 2025-07-31

**Authors:** Swamy R. Adapa, Karen Hendrix, Aditya Upadhyay, Subhajeet Dutta, Andrea Vianello, Gregory O’Corry-Crowe, Jorge Monroy, Tatiana Ferrer, Elizabeth Remily-Wood, Gloria C. Ferreira, Michael Decker, Robert H. Tykot, Sucheta Tripathy, Rays H. Y. Jiang

**Affiliations:** 1USF Genomics, Global Health Infectious Disease Research Center (GHIDR), Global Health, College of Public Health, University of South Florida, Tampa, FL 33612, USA; 2Near Eastern Archaeology Foundation, University of Sydney, Sydney, NSW 2006, Australia; 3Structural Biology and Bioinformatics Division, CSIR-Indian Institute of Chemical Biology, Kolkata 700032, India; 4Academy of Scientific and Innovative Research (AcSIR), Ghaziabad 201002, India; 5Department of Anthropology, University of South Florida, Tampa, FL 33612, USA; 6Harbor Branch Oceanographic Institute, Florida Atlantic University, Fort Pierce, FL 33612, USA; 7Department of Molecular Medicine, Morsani College of Medicine, University of South Florida, Tampa, FL 33612, USA; 8Department of Chemistry, College of Arts and Sciences, University of South Florida, Tampa, FL 33620, USA; 9Department of History, University of South Florida, Tampa, FL 33612, USA

**Keywords:** First Pandemic, Justinianic Plague, Yersinia pestis, ancient DNA, Jerash, Plague

## Abstract

**Background/Objectives:** The Plague of Justinian marked the beginning of the First Pandemic (541–750 CE), yet no genomic evidence of *Yersinia pestis* has previously been recovered from the Eastern Mediterranean, where the outbreak was first recorded. This study aimed to determine whether *Y. pestis* was present in a mid-6th to early 7th century mass grave in Jerash, Jordan, and to characterize its genome within the broader context of First Pandemic strains. **Methods:** We analyzed samples from multiple individuals recovered from the Jerash mass grave. Initial screening for potential pathogen presence was conducted using proteomics. Select samples were subjected to ancient DNA extraction and whole genome sequencing. Comparative genomic and phylogenetic analyses were conducted to assess strain identity and evolutionary placement. **Results:** Genomic sequencing recovered *Y. pestis* DNA from five individuals, revealing highly similar genomes. All strains clustered tightly with other First Pandemic lineages but were notably recovered from a region geographically close to the pandemic’s historical epicenter for the first time. The near-identical genomes across diverse individuals suggest an outbreak of a single circulating lineage at the time of this outbreak. **Conclusions:** This study provides the first genomic evidence of *Y. pestis* in the Eastern Mediterranean during the First Pandemic, linking archaeological findings with pathogen genomics near the origin point of the Plague of Justinian. **Summary Sentence**: Genomic evidence links *Y. pestis* to the First Pandemic in an ancient city.

## 1. Introduction

Three plague pandemics have been recorded in history over Western Eurasia and portions of Africa. The First Pandemic, the so-called Plague of Justinian began, according to historical sources [1,2], in 541 CE and lasted until ca. 750 CE. The disease first appeared in the historical record at Pelusium (present day Tell el-Farama) in Northern Egypt, then part of the Eastern Roman (Byzantine) empire, and was believed by contemporaries to have begun in East Africa or South Arabia, whence it rapidly spread throughout the Mediterranean and Middle East [3] (Figure 1A). The Second Pandemic, starting with the Black Death from 1346–1353 CE, is much better documented historically and resulted in a death rate of about 65% of the European population, or 80 million people, and at least 25 million in Asia and Africa [4,5]. The Third Pandemic, started in China, has resulted in the deaths of over 15 million people and continues today, with plague remaining endemic in numerous countries worldwide and annual outbreaks reported [5].

The plague is caused by *Y. pestis*, a Gram-negative zoonotic coccobacillus. Since plague leaves no bone lesions or other distinguishing skeletal features, molecular evidence of plague is required to confirm infection in human remains and can be obtained by extraction and molecular analysis of highly vascularized tissue like dental pulp [6]. The application of aDNA protocols to the pulverulent dental pulp led to the first molecular identification of *Y. pestis* in individual skeletal remains of victims of the Second Pandemic and opened the study of the paleomicrobiology of the disease [6].

However, to date, no *Y. pestis* genetic material has been recovered in the Eastern Mediterranean in geographic and temporal proximity to the epicenter of the First Pandemic. Currently, a few sites in Central and Northwestern Europe have yielded skeletal remains and ancient DNA linked to the Plague of Justinian time period; e.g., two in Germany, one in Spain, three in France, and one in England [7]. These sites are far from the reported epicenter of the Plague of Justinian, with Bavaria in Germany lying 2500 km distant and Edix Hill cemetery in England more than 3500 km distant from the epicenter of Pelusium.

There is also a lack of archaeological evidence of mass graves within the Eastern Mediterranean, where the plague emerged and likely had more severe effects. While mass graves, or ‘plague pits’ as they are sometimes called, have been found in considerable number relating to the Second Pandemic of the 14th century CE, none have thus far been identified as relating to the First Pandemic.

Here, we report archaeological and genetic findings of *Y. pestis* in multiple individuals from a mass grave approximately 550–660 CE in Jerash in Jordan, the ancient city of Gerasa, which lies 50 km north of Amman. We therefore present the first archeological and molecular evidence of *Y. pestis* linked to the time of the Plague of Justinian within the confines of the Roman Empire and geographically proximate to the historically attested epicenter: Jerash is just 330 km from Pelusium.

### Archaeological Context—Jerash (Ancient Gerasa)

Jerash was an important city during the Greco-Roman and Byzantine periods and is considered one of the largest and most well-preserved sites of Roman architecture in the world outside Italy [8]. Among the many accoutrements of Roman rule was the installation of a hippodrome (chariot racetrack or Roman Circus) outside the city center. The Jerash Hippodrome was built in the mid-2nd century CE and ceased to exist as a chariot racing arena in the first quarter of the 4th century CE [9]. (*An ideal reconstruction of the 2nd century CE Roman city of Gerasa*, *located in present-day Jerash*, *Jordan*, *can be viewed in JR. CASALS’ rendering work*.)

After the Hippodrome ceased as a chariot racing venue, the storage chambers under the seating were part of structures in the Hippodrome to be repurposed for light industry, including ceramic workshops dating from the Late Roman period that continued into the Early Byzantine period (mid-4th century CE) [10]. A compact layer of misfired and unfired Late Byzantine (the second half of the 6th century CE) pottery, including fragments of ‘Jerash Bowls’, sealed these installations. The entire building was abandoned around the latter part of the 6th century CE or beginning of the 7th century CE [11].

The 1993 excavations of the W2 and W3 chambers (Figure 1B,C) discovered human skeletons immediately above the dumps of the misfired Byzantine pottery, which had been crushed underneath and within the tumbled seating blocks [11,12] (Figure 2A). Our samples were obtained from these interments, which held approximately 150 adults and 80 sub-adults (neonates, infants, and children) [13,14].

Since the initial examination of the mass grave, access to the collection has not been possible, and only a small collection of dentition was available for selection and analysis. From this collection, thirty-seven loose molars and two ribs were chosen for further examination based on their overall completeness, lack of caries, and other dental disease.

## 2. Methods and Results

### 2.1. Proteomic Results Showed Potential Presence of the Plague Pathogen

As a preliminary screening tool, ancient proteomic analysis was conducted using a combination of mass spectrometry-based techniques. Protein extraction from well-preserved dental samples (n = 10) was performed under controlled laboratory conditions to minimize contamination. Extracted proteins were then subjected to liquid chromatography–mass spectrometry (LC-MS/MS) for identification and quantification. Spectral data were processed, with a focus on identifying pathogenic proteins. (Detailed protocols are in the Appendix A.) We obtained 25–368 unique human peptides from the samples, with 4 samples having at least 3 *Y. pestis* peptides present based on MaxQuant v2.4 (Appendix A) (Figure 2B). We found that the teeth were well-preserved in terms of structural integrity and hardness, allowing us to generate proteomics data. This provides evidence that the Jerash biomaterial was suitable for further genetic and genomic analysis and indicates the possible presence of *Y. pestis* in the burial sites. These quality control measures, along with the detection of multiple pathogen-specific peptides across different individuals, increase confidence in the proteomic signal. Importantly, blank extractions and reagent-only controls processed in parallel did not yield any *Y. pestis* peptide hits, supporting the authenticity of the findings and ruling out laboratory or reagent-derived contamination.

### 2.2. Whole Genome Sequences Revealed the Pandemic Pathogen Genome

Dentition from 5 individuals (represented by 8 teeth) were carefully selected for aDNA analysis based on criteria such as preservation status and bone density (Figure 2C). Prior to extraction, external surfaces of the teeth were cleaned with a mild bleach solution and physically abraded to remove potential contaminants. The aDNA extraction was performed in a dedicated clean room environment following established protocols optimized for ancient samples [15,16]. Briefly, tooth samples were stored in airtight containers in a secure room separate from the Ancient DNA Facility. All experiments were conducted in a certified aDNA clean room complex with stringent air quality controls. Lab personnel followed strict protocols: showering before entry, donning protective suits, and wearing double sterile gloves, face masks, and eye protection within a sterile zone. Work was performed under fluorescent and UV lamps in dedicated benchtop hoods. Each tooth was processed individually using a handheld Dremel^®^ tool with sterile attachments. Prior to use, all tools and surfaces were sterilized with bleach and UV light.

Tooth powder was collected from pulp, dentine, and cementum using sterilized equipment. In cases where pulp and dentine could not be distinguished, a mixed sample was collected. Cementum was crushed separately. Samples were weighed, and DNA extraction required 30–50 mg of powdered material. All equipment was sanitized and irradiated after each use to prevent contamination. DNA extraction used the PrepFiler™ BTA Forensic DNA Extraction kit in a separate chamber under similar sterile conditions. Extracted genomic DNA was eluted in PrepFiler™ Elution Buffer and stored at −20 °C. (Detailed protocols are in the Appendix A.) Negative extraction controls were conducted simultaneously with sample extractions and total DNA was quantified using a Qubit Flex Fluorometer.

To obtain and authenticate aDNA, library preparation for whole genome sequencing was carried out using a modified protocol for aDNA, incorporating steps to mitigate potential DNA damage and fragmentation inherent to ancient samples. Library amplification was conducted with a limited number of cycles to prevent over-amplification of damaged DNA molecules. Whole genome sequencing was carried out on an Illumina sequencing platform using paired-end sequencing reads. We used three independent aDNA authentication methods to verify that the DNA was from the Jerash material: (1) obtaining short fragments (<100 bp) due to ancient DNA degradation, (2) sequencing untreated (not gap-filled) DNA yielded gapped results, and (3) computational validation of whole genome comparative analysis showing ancient lineages.

We obtained large number of total reads, coverage ranges from 1.25x to 35x of the *Y. pestis* genome from 8 samples derived from 5 individuals, and the *Y. pestis* genomes were constructed. The Jerash isolates showed an average nucleotide identity (ANI) of 0.9999 with whole genome variants comparisons. This assessment was conducted using two independent ANI pipelines (Appendix A): one computational pipeline comparing pruned genomic variants (Pipeline A) and the other (Pipeline B) using the complete set of genomic information. The pruned genomic variants refer to a curated set of high-confidence single nucleotide polymorphisms (SNPs) derived after quality control steps, including duplicate read removal and damage pattern filtering, ensuring that only reliable genomic differences were used for ANI comparisons. Both pipelines yielded highly similar results. Thus, the pathogen isolate’s genomes from the different samples all represent the same isolate with highly similar genomes (Figure 3A and Appendix A) despite that the human host isotope signatures differ from each other. Our data shows that the demographically diverse victims bear the same *Y. pestis* isolates in this mass grave site.

### 2.3. Virulence Factor Analysis

For virulence factor analysis, we first removed low quality reads with BBMap and used the BWA-MEM algorithm to generate aligned BAM files. We found that diverse strains spanning from prehistory to the pandemics possess various virulence factors, suggesting that virulence evolution may have begun early in Neolithic/Bronze Age human societies (Figure 3B). Regarding the flea transmission-related gene *Ymt*, the prehistorical strain [17,18] of RT5/RT6 possesses this gene as previously reported. Both the First Pandemic and Second Pandemic strains (including the Black Death strains) contain *Ymt*. Our results show that the Jerash mass grave strain harbors important virulence factor genes [19], such as *Ymt*, *Pla*, and *F1 capsule genes*, which are associated with the transmission and pathology of *Y. pestis*.

### 2.4. Tracking Potential Transmission of the First Pandemic

We have implemented NextStrain, similar to Eaton et al. [20]. We performed maximum likelihood (ML) phylogenetic analysis [21] with the whole genome genetic variants (Figure 4). The best fit model was chosen according to Bayesian information criterion (BIC) [22]. Tree topology bootstrapping was performed for 1000 replicates.

We demonstrate that the Jerash strain aligns closely with strains recovered from the same period as the First Pandemic (541–750 CE), forming a highly supported cluster. Significantly, the Jerash strain represents the first genome recovered within the Eastern Mediterranean region, which historically served as the epicenter of the initial global pandemic.

The Tian Shan Hun strain forms a closely supported lineage with First Pandemic strains, consistent with its geographical location in Central/Western Asia, in the Tian Shan mountain (天山) at the edge of the Eurasian steppes. Jerash, as the nearest large urban center so far examined in proximity to the likely area of the plague origin, supports the hypothesis of introduction from the Eurasian steppes.

## 3. Discussion

This study set out to investigate the presence of *Y. pestis* in human remains from a mass grave at the ancient archaeological site of Jerash, Jordan. Through combined proteomic screening and high-confidence ancient DNA sequencing, we successfully identified *Y. pestis* genomes from multiple individuals. These represent the first pathogen genomes recovered from this region and period, thereby bridging a major geographic and temporal gap in our understanding of the First Pandemic’s genetic footprint [2,23,24].

There is a possibility of concurrent infections by multiple pathogens within the same host, especially in chronic diseases such as tuberculosis (TB) and leprosy [25,26,27]. Further investigation is warranted to specifically explore the interplay among chronic bacterial and viral diseases. Addressing this multiplicity question could provide deeper insights into the complex dynamics of co-infections and their implications for large outbreaks.

The genomic analysis revealed that all *Y. pestis*-positive individuals shared highly similar isolates, indicating that a single strain was likely responsible for the outbreak at Jerash. This homogeneity, seen across individuals of diverse age and isotopic background (unpublished results), points to a rapid and localized transmission event. Such findings align with textual accounts of sudden mass mortality and support the hypothesis that early documented pandemic waves were marked by explosive outbreaks [28,29] rather than long, smoldering transmission chains. Importantly, the presence of a uniform strain in an urban population provides molecular evidence for the scale and speed of infection in densely settled communities.

Urban centers like Jerash were hubs of trade, administration, and population movement within the Late Roman and Early Byzantine world [9,24,30]. Their civic infrastructure—including aqueducts, baths, granaries, and amphitheaters—facilitated not only the flow of goods and people but also, inadvertently, pathogens. These environments created ideal conditions for the rapid spread of flea-borne diseases, particularly within crowded and unsanitary quarters of aging cities. Our study shows how the very fabric of Late Roman urbanism—its interconnectedness, mobility, and centralization—may have made cities like Jerash especially vulnerable to epidemic incursion.

Equally important is the archaeological context of the burial. The repurposing of the disused Roman hippodrome at Jerash for mass interment offers rare physical evidence of crisis-era responses to public health catastrophe. This adaptation of civic architecture to accommodate emergency burial needs reflects the breakdown of traditional mortuary practices and hints at the social dislocation triggered by the outbreak. That such a prominent structure was transformed into a cemetery [9,10,12,31] suggests a tipping point, where infrastructure intended for entertainment or assembly was reappropriated for collective mourning and pragmatic disposal of the dead.

Finally, our findings raise broader questions about the interplay between zoonotic reservoirs and urban pandemics. The repeated re-emergence of *Y. pestis* across millennia (17–20) reflects a pathogen capable of both environmental persistence and episodic amplification. As we expand the genomic record of past outbreaks, including sites like Jerash, we move closer to reconstructing a fuller picture of the plague’s deep history—not as a series of isolated tragedies but as a part of a recurring ecological phenomenon shaped by human behavior, urban form, and environmental entanglements. The case of Jerash highlights how urbanization itself—its density, infrastructure, and connectivity—was both the engine of imperial prosperity and the Achilles’ heel of public health, amplifying vulnerability to pandemic disease. Understanding how ancient cities absorbed, responded to, and were transformed by epidemic shocks remains central to the story of plague, past and present.

## 4. Conclusions

Our interdisciplinary investigation, employing archaeological, proteomic, genetic, and genomic techniques, provides important insights into this First Pandemic (541–750 CE). The analysis of our samples from the excavation of a mass grave site in Jerash, Jordan, offers a critical window into the development of plague pandemics. Our genetic analysis links the pandemic to *Y. pestis*, thereby confirming historical accounts of the Plague of Justinian for the first time. The phylogenetic analysis supports its initiation from the Eastern steppes.

## 5. Limitations

The limitation of our study is the potential preservation bias, as it relies on a relatively small number of well-preserved ancient samples, which may not fully capture the genetic diversity of plague strains. Additionally, interpreting historical epidemiological patterns is constrained by the available historical records, which may not encompass all aspects of past pandemics.

## Figures and Tables

**Figure 1 genes-16-00926-f001:**
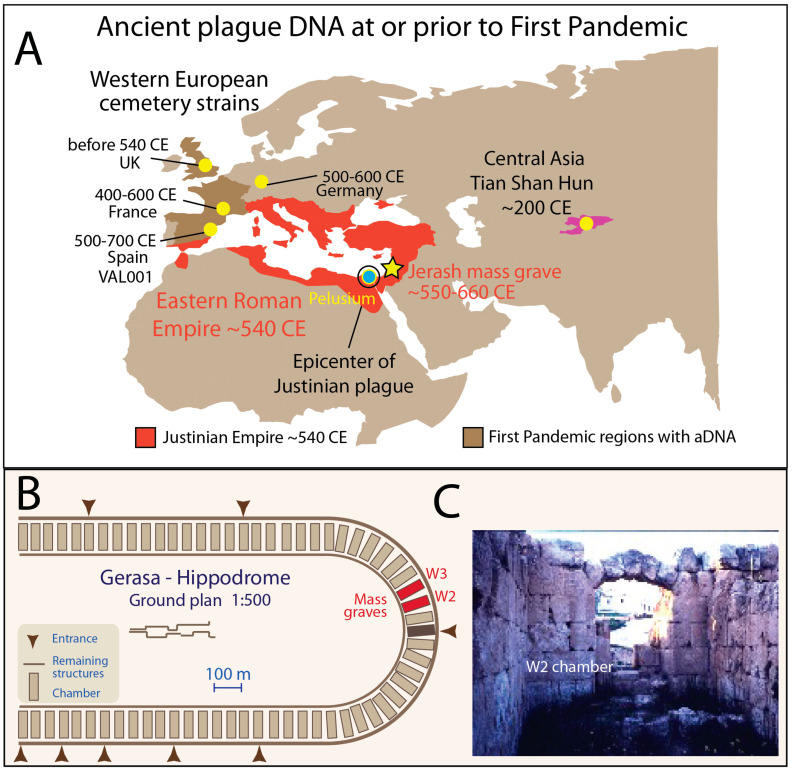
Jerash mass grave site in the context of the First Pandemic. (**A**). Regions of aDNA isolated to date prior or at the First Pandemic are marked in dark brown (around First Pandemic). The Jerash site is marked by a star. The Eastern Roman Empire (indicated in dark red) historically documented the plague spreading from North Egypt and the Near East through Mediterranean regions. Yellow dots in dark brown regions denote the current available genetic evidence of *Y. pestis* recovered from Western European cemeteries. The single Tian Shan Hun Strain (yellow dot) in Central/Western Asia is hypothesized to be one of the early strains present prior to the onset of the Plague of Justinian. (**B**). Jerash (Gerasa) hippodrome ground plan. The schematic of the ground plan was redrawn based on the illustration of Anton Ostrasz. The mass graves were uncovered in the abandoned Roman hippodrome chambers W2 and W3. (**C**). One of the chambers (W2) where the bodies were recovered.

**Figure 2 genes-16-00926-f002:**
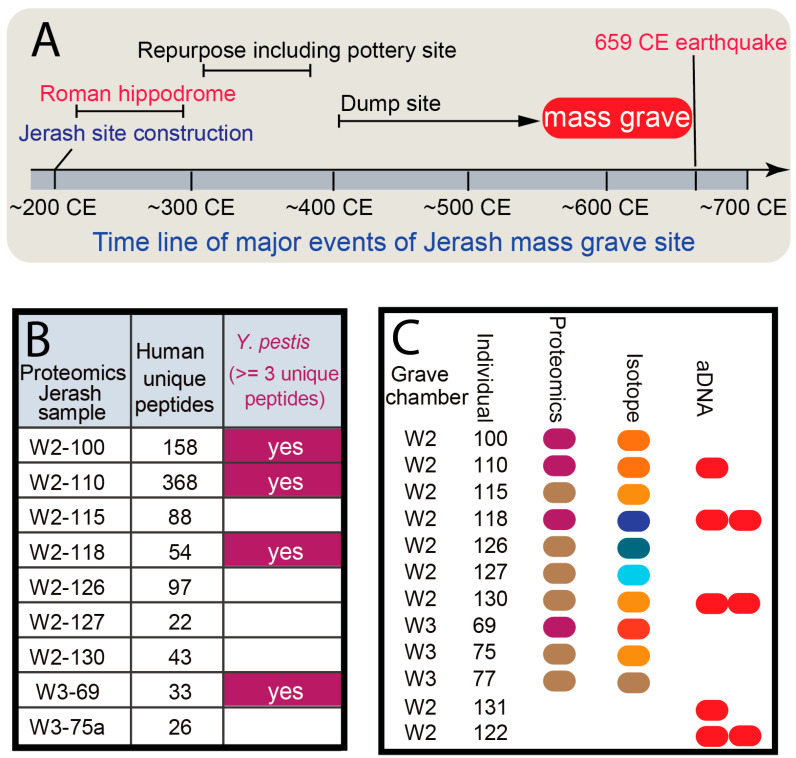
Archaeological, proteomic, and genetic evidence of an ancient epidemic in Jerash. (**A**). Archaeological excavation records narrow the mass grave timeframe. The abandoned Roman Hippodrome site was used as a mass grave site around the time of the Plague of Justinian. (**B**). Proteomics analysis identified potential peptides from *Y. pestis*. At least three unique pathogen peptides passing quality control in experimental and analysis protocols were required. (**C**). Summary of samples that yielded proteomics, stable isotopes of oxygen, and ancient DNA (aDNA) results. *Y. pestis*-positive proteomics samples are colored in purple. For three individuals, two independent aDNA samples were generated from two teeth of the same individual.

**Figure 3 genes-16-00926-f003:**
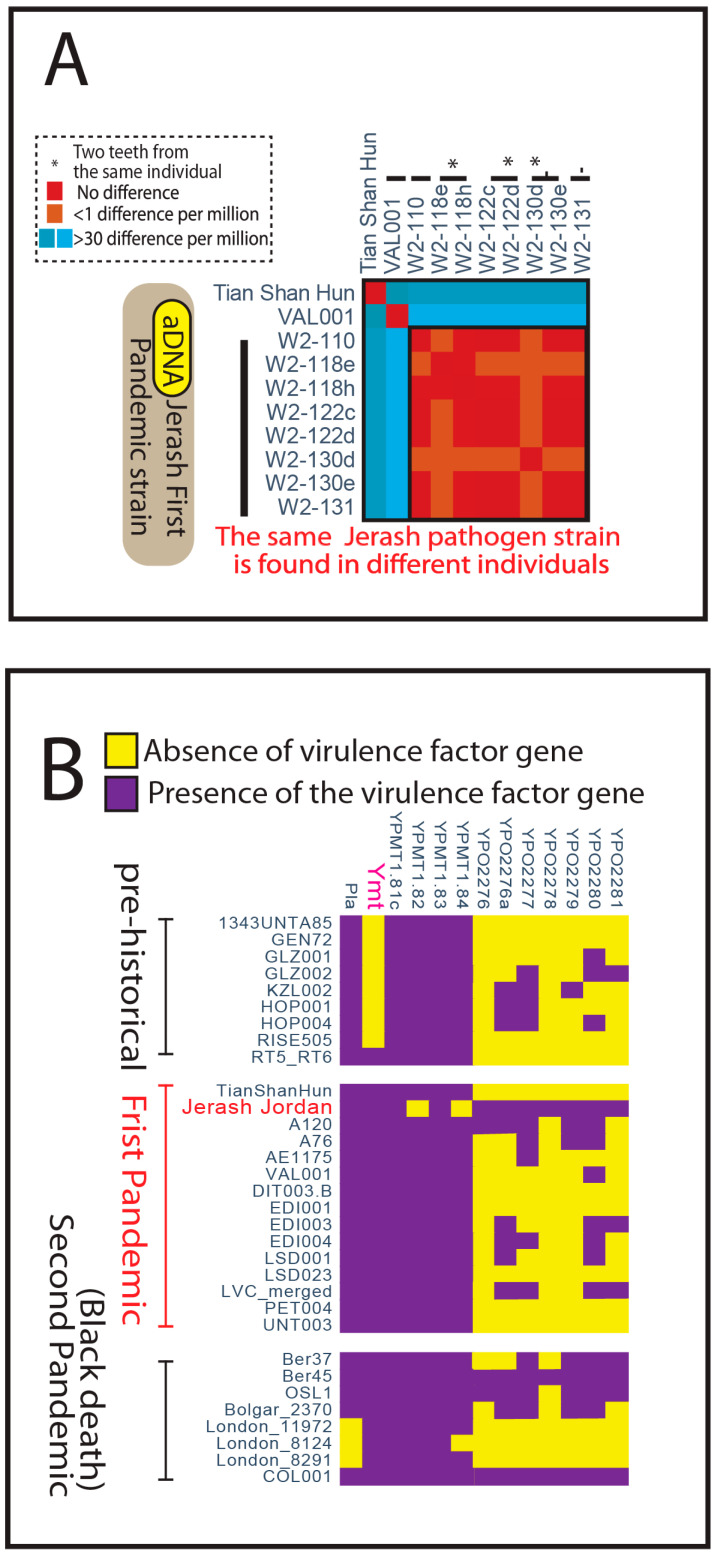
Genome reconstruction and analysis of *Y. pestis* in Jerash. (**A**). Whole genome capture recovered the same Jerash *Y. pestis* strain from different victims of the mass grave. Average nucleotide identity (ANI) analysis revealed differences of equal to or less than 1 base per million among Jerash isolates, contrasting with over 30 bases per million differences in other First Pandemic-related strains like VAL001 and Tian Shan Hun. Sequencing of two teeth from each of three individuals showed no significant differences, indicated by *. (**B**). The Jerash strain contains many essential virulence factors for causing an epidemic. A set of virulence factor genes from prehistorical, First Pandemic, and Second Pandemic (Black Death) strains are shown. Many prehistorical strains lack the flea transmission-related gene *Ymt*, while RT5/RT6 does possess the *Ymt* gene. Both Tian Shan Hun and Jerash strains possess the *Ymt* gene.

**Figure 4 genes-16-00926-f004:**
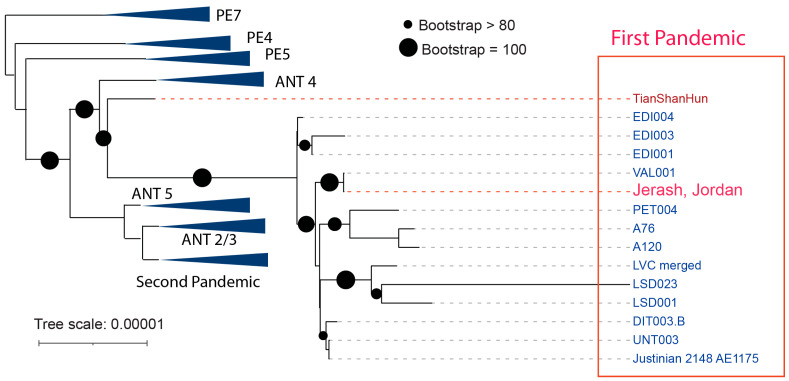
Phylogenetic confirmation of First Pandemic plague strains. The maximum likelihood phylogenetic reconstruction of *Y. pestis* strains from the First Pandemic. Tree topology bootstrapping was performed with 1000 replicates to assess the robustness of the inferred relationships. The Jerash strain groups with other First Pandemic strains and their branch lengths are represented. Dots along the branches indicate bootstrap support values greater than 80%, providing strong confidence in those relationships. Collapsed branches are shown with triangles. The Jerash strain is highlighted in red, while the Tian Shan Hun strain is shown in dark red, emphasizing their positions within the phylogeny relative to other strains from the same period.

## Data Availability

Data needed to evaluate the conclusions in the paper are available in the paper, Appendix A, and the NCBI Sequence Read Archive (SRA) data archive (https://www.ncbi.nlm.nih.gov/sra/PRJNA1303691; accessed on 9 August 2025). Accession to cite: PRJNA1303691; Temporary Submission ID: SUB15508200; Release date: 9 August 2025.

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
