# Peer review of "Genetic Evidence of Yersinia pestis from the First Pandemic"

_genes, 2025, doi:10.3390/genes16080926_

Round 1
Reviewer 1 Report
Comments and Suggestions for Authors
The manuscript deals with the characterization of Y. pestis from the first pandemic using aDNA techniques and MS analyses.
The manuscript is very interesting, and the introduction is well-written, informative, and comprehensive. The authors conclude that all the human samples they analysed shared the same pathogen.
The discussion, in contrast, is too concise, so I recommend a thorough revision to expand it and better discuss the results and their implications, in the context of the literature on the topic.
Reviewer 2 Report
Comments and Suggestions for Authors
Adapa et al presents genetic evidence of Yersinia pestis from the first pandemic. Overall, the question addressed is interesting, but authors need to clarify their methods.
While the detection of Y. pestis peptides is suggestive, proteomics alone is not definitive for ancient pathogen identification due to issues of contamination and peptide overlap with related species. Please clarify how the proteomic hits were validated and what steps were taken to rule out false positives or background noise, especially given the high homology of Yersinia species proteins.
Were any blank extractions or reagent-only controls included during the proteomics workflow? Ancient proteomics is highly susceptible to modern contamination, and this should be explicitly addressed.
The Discussion section is relatively short and lacks a comprehensive synthesis of the study’s key findings. Much of the current Discussion overlaps with the Limitations section, rather than providing a critical interpretation of the results, their implications for the geographic and genomic understanding of Y. pestis spread during the Plague, and how these findings integrate with existing archaeological or textual evidence.
The main manuscript refers to detailed methods being available in the Supplementary Information; however, I was unable to locate or access a complete Supplementary Methods section.
Line 173: Please briefly define pruned genomic variants for readers unfamiliar with the method.
Round 2
Reviewer 1 Report
Comments and Suggestions for Authors
The revised version of the manuscript includes further details and improvements, but still lacks of an adequate discussion section, despite authors' claims in their response to reviewers. I'd strongly encourage authors to extensively improve the discussion.
Reviewer 2 Report
Comments and Suggestions for Authors
Authors have addressed most of my concerns. Discussion section is still short, but is much better now.
Overall, plagiarism is 43%. Authors should consider to lower it before the paper can be accepted.
